# Assessment of Resistance Mechanisms to Fall Armyworm, *Spodoptera frugiperda* in Tropical Maize Inbred Lines

Ovide Nuambote-Yobila [1,2,3,4], Anani Y. Bruce [5], Gerphas Ogola Okuku [5], Charles Marangu [5], Dan Makumbi [5], Yoseph Beyene [5], Nzola-Meso Mahungu [1], Boddupalli Maruthi Prasanna [5], Frédéric Marion-Poll [3,4] and Paul-André Calatayud [2,3,*]

[1] Académie de Recherche, d'Innovation et de Formation Professionnelle en Agriculture pour le Développement (ARIFPAD), CPAID, Université Kongo, Mbanza-Ngungu B.P. 202, Democratic Republic of the Congo
[2] International Centre of Insect Physiology and Ecology (icipe), Nairobi P.O. Box 30772-00100, Kenya
[3] Université Paris-Saclay, CNRS, IRD, UMR Évolution, Génomes, Comportement et Écologie, 91198 Gif-sur-Yvette, France
[4] Université Paris-Saclay, AgroParisTech, CEDEX 05, 75231 Paris, France
[5] International Maize and Wheat Improvement Centre (CIMMYT), Nairobi P.O. Box 1041-00621, Kenya
* Correspondence: paul-andre.calatayud@universite-paris-saclay.fr

**Abstract:** The fall armyworm (FAW), *Spodoptera frugiperda*, a pest of maize native to the Americas first reported in West and Central Africa in 2016, severely threatens maize production and food security in Sub-Saharan Africa. Native genetic resistance is one of the best methods of control of insect pests as it is contained in the seed making it more amenable for use by farmers compared to other interventions and it is also compatible with other integrated pest management (IPM) options. An intensive screening against FAW was carried out by artificial infestation in greenhouse conditions in Kenya between 2017 and 2018 on about 3000 inbred lines available in the germplasm collection of the International Maize and Wheat Improvement Center (CIMMYT). Among these lines, only four showed to be resistant to FAW, but the mechanisms of resistance are not yet known. The objective of this study was to determine the resistance mechanisms specifically non-preference and antibiosis to *S. frugiperda* in these four selected resistant inbred lines. The studies were conducted under laboratory and net house conditions in Kenya from April 2020 to November 2021. Non-preference was assessed estimating the feeding preference by counting the number of FAW neonates found on each leaf portion, silk portion and grain using binary and multiple choice methods under laboratory conditions, while antibiosis was assessed through the relative growth rate (RGR) and developmental time of FAW larvae on leaves, silks and grains under both laboratory and net house conditions. Among the four resistant maize inbred lines tested, two, namely CML71 and CKSBL10008, exhibited the highest level of antibiosis resistance on leaves. Under laboratory conditions, the larval RGR reduced from 13 mg/d on the most susceptible line to 8 mg/d on CML71. CML71 also showed a good non-preference on leaves compared to other tested lines. Only 6% of neonates choose to feed on CML71 whereas more than 10% choose to feed on the other lines (and 15% on the most susceptible) in multiple choice tests. The non-preference for feeding and lower RGR of larvae on CML71 suggest a biochemical involvement resistance to FAW. Through this study, CML71 is revealed as a highly promising line for use in breeding for native genetic resistance to FAW in tropical maize.

**Keywords:** native resistance; antibiosis; antixenosis or non-preference; maize inbred lines; FAW; Africa

## 1. Introduction

Maize (*Zea mays* L.) is the most important food crop in terms of production volume and provides more than 20% of food calories for the human population in sub-Saharan Africa [1]. The fall armyworm (FAW), *Spodoptera frugiperda* (JE Smith) (Lepidoptera: Noctuidae), a pest of maize native to the Americas first reported in West and Central Africa in

2016 [2], is severely threatening food security in sub-Saharan Africa. Day et al. [3] estimated maize production losses due to FAW at between USD 2.481 and 6.187 million per annum. In the African context where most maize farmers are smallholders with limited access to knowledge and adequate inputs to manage this pest [4], host plant resistance is one of the most effective means of control and is compatible with other integrated pest management strategies [5]. Farmers in African countries are resource-constrained smallholders and already face problems in effectively controlling FAW using insecticides [6]. Environmentally safer pesticides are usually costlier than toxic pesticides. In addition, most of the smallholders are not aware of pesticide risk management, and do not have proper personal protective equipment. Therefore, development and deployment of insect-resistant maize will provide a major boost to maize production in sub-Saharan Africa.

Development of insect-resistant maize germplasm is a difficult process, but some progress has been made over the years. In the Americas, some maize inbred lines resistant to FAW have been developed such as Mp708, Mp78:518 [7,8] and Mp716 [9]. The International Maize and Wheat Improvement Center (CIMMYT) embarked on the development of maize germplasm with insect resistance beginning in the 1980s resulting in multiple borer-resistant (MBR) and multiple insect resistance tropical (MIRT) populations, and more recently, germplasm was developed under the Insect Resistant Maize for Africa (IRMA) initiative [6,10]. Between 2018 and 2019, CIMMYT carried out intensive screening of maize inbred lines under artificial FAW infestation in Kenya and identified several promising FAW-resistant inbred lines that had low leaf and ear damage [11]. Among the best inbred lines in terms of FAW resistance were CML71, CML125 and CML370 which were derived from the multiple borer-resistant (MBR) population, and CKSBL10008, a resistant line to stem borer developed through the IRMA Project. However, the resistance mechanisms to FAW larvae in these lines is not yet known. Painter [12] defined three basic resistant mechanisms in plants towards herbivorous pests, namely antixenosis (or non-preference), antibiosis and tolerance.

The objective of this study was to determine the antibiosis and antixenosis to FAW in selected inbred lines CML71, CML125, CML370 and CKSBL10008 under laboratory and net house conditions.

## 2. Materials and Methods

### 2.1. Plant Materials

The four selected FAW-resistant maize (CML71 CML125, CML370, and CKSBL10008) were used in this study. In addition, a known FAW-resistant maize inbred line from USDA (Mp716) was used as a positive check; and two most susceptible (CML444 and CKSBL10025) inbred lines highlighted from the intensive screening by CIMMYT were used as negative checks. Seed of the inbred was obtained from the CIMMYT maize breeding program in Kenya.

### 2.2. FAW Larvae Colony

A colony of FAW was established from samples collected in the field. Twice a year, about 200 larvae and pupae of FAW were collected from maize fields of Kiboko (02°21′ S, 037°70′ E, 945 m.a.s.l.) and Machakos (01°57′ S, 027°25′ E, 1568 m.a.s.l.) in the *Eastern* region of Kenya. The colonies were maintained at 25 ± 1 °C, RH of 75% ± 5 and a photoperiod of 12:12 (L:D) h., at the CIMMYT/Kenya Agricultural and Livestock Research Organization (KALRO) insectary at Katumani, Machakos, on an artificial diet using a protocol optimized by CIMMYT [11]. First-instar larvae (neonates) of third laboratory-reared generation were used for all experiments.

### 2.3. Plants Used in the Laboratory Experiment

Plants for the laboratory experiments were grown in plastic pots (26.2 cm high × 25 cm in diameter) filled with a mixture of soil and farmyard manure at the rate of 3/4 + 1/4 in a greenhouse at KALRO Katumani (30 °C, 50 % RH, 12L: 12D). One seed was planted

per pot. Five days after plant emergence, 10 g of diammonium phosphate (DAP) were applied to each pot. Irrigation was carried out when needed. For the leaf bioassay, leaves of plants at the V5 maize's growth stage were infested with 5 neonates of FAW each, as described by [11]. In addition, ears at the R1 silking stage (7 days after silking) and at R3 milk stage were used for all laboratory experiments [13].

### 2.4. Plants Used in the Net House Experiment

The net house experiment trial was conducted at the FAW Screening Facility at Kiboko. The plants were grown on the ground in complete randomized blocks with three repetitions, with plant spacing of 0.75 m between the rows and 0.25 m within rows, in a net house of 326.25 m² (15 m × 21.75 m × 5 m) (see Figure 1) at KALRO, Kiboko Research Center (Kenya) at 28 °C, 48% RH, 12 L:12 D. One week before planting, Emamectin benzoate (19 g/L) insecticide was sprayed on the ground and walls of the net house against insects that might interact with the experiment. Two seeds were sown per hill and thinning was carried out at 2 weeks after planting, leaving one plant per hill. Approximately 10 g of diammonium phosphate (DAP) was applied to each hole five days after emergence of plants; 10 g of calcium ammonium nitrate (CAN) was added to each hole at the tasseling stage of the plant for top dressing. Manual weeding was carried out when necessary. The plants were regularly irrigated. Plants were used at the V5, R1 silking (7 days after silking) and at R3 milk stage.

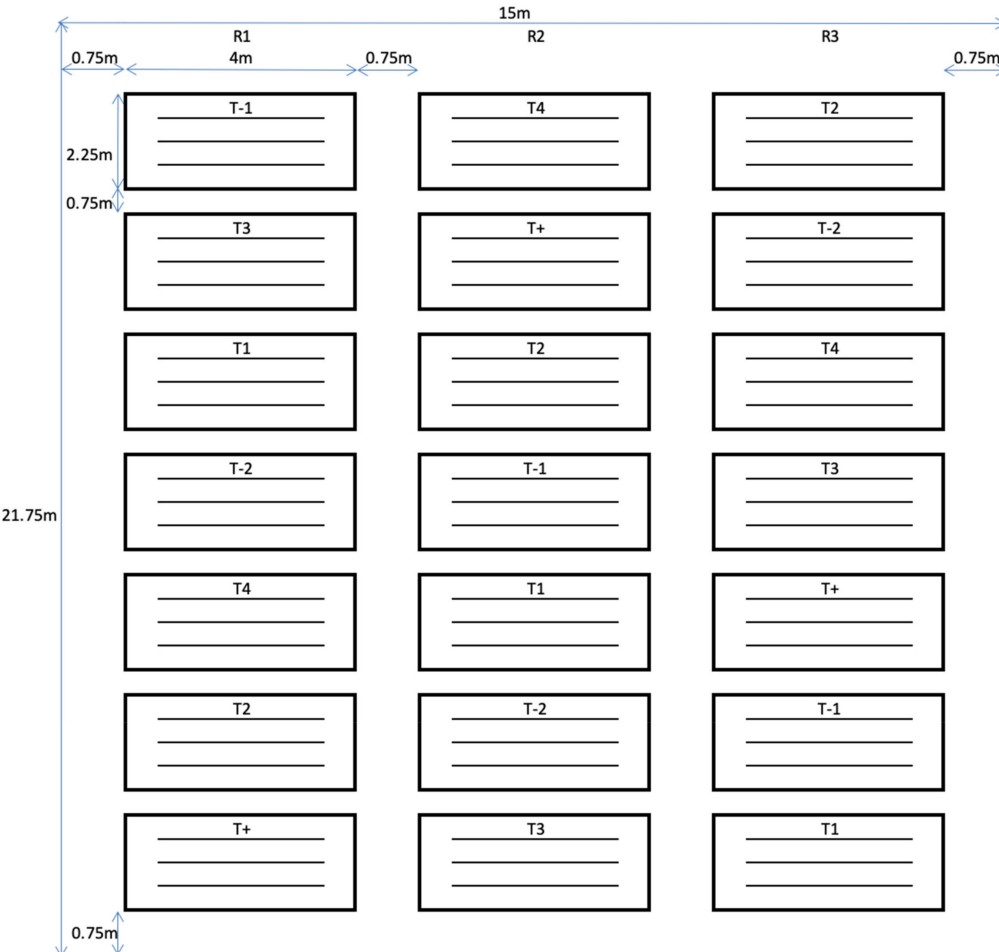

**Figure 1.** Experimental design of the experiment to assess the antibiosis of lines infested by larvae of *Spodoptera frugiperda* on leaves, on silks and ears (see Section 2.6). R1: 1st block; R2: 2nd block; R3: 3rd block; T-1: Plot of CML444; T-2: Plot of CKSBL10025; T +: Plot of Mp716; T1: Plot of CML71; T2: Plot of CML125; T3: Plot of CML370; T4: Plot of CKSBL10008.

### 2.5. Antibiosis Assessment under Laboratory Conditions

To assess antibiosis among the inbred lines, young leaves were collected every day from V5 to V10 growth stages from each potted plant of each inbred line and fed to each larva until the pupal stage. Three leaf portions, each approximately 5 cm long and 3 cm wide, and a single neonate were placed in a 25 mL plastic vial (8.3 cm long × 2.3 cm diameter) and closed with cotton wool to prevent the escape of larvae but allow air circulation. Similar to [14], a single neonate was used since FAW is known to be cannibalistic when the larvae are reaching the older stages [15]. Vials and leaf portions were changed after 2 days at the beginning of the experiment and then daily from the fourth larval instar up to the pupal stage. Each pupa was kept in the 25 mL vial until adult emergence. From each inbred line, a freshly emerged adult male and female were paired in a jar of 20 cm high, and 11 cm diameter sealed with a paper towel and a perforated plastic cap. In each jar, a piece of cotton wool soaked in a 15% sucrose solution was added to feed the moths, together with two portions of leaves of 15 to 20 cm each from the corresponding inbred line for oviposition. The sucrose-soaked cotton wool was renewed after 2 days while the leaf portions used for oviposition were renewed daily. Batches of laid eggs were collected daily and stored in a vial closed with cotton wool for counts of eggs and time of hatching. To estimate the sex ratio, a minimum of 200 neonate larvae (a minimum number giving a chance to represent the sex ratio of a progeny) from the total hatched eggs of each female from each inbred line were randomly reared on an artificial diet until pupation using the protocol optimized by CIMMYT in Kenya [11]. The sex ratio was calculated as the number of females/the total number of adults.

The following parameters were recorded from insects from each inbred line: relative growth rates (RGR in mg/d) after 10 days of feeding, larval development time (days) from neonate to pupa, percentage of larval survival, weight (mg) of male and female pupae, female pre-oviposition period (days) after mating (in days), the female oviposition period, total number of eggs laid per female, percentage of eggs hatched (i.e., related to females' fertility) and the sex ratio. The percentage of larval survival is calculated as the number of alive larvae/total number of larvae used for the infestation. The percentage of eggs hatched is calculated as the number of eggs hatched/total number of eggs laid by female.

The RGR value of a larva on each plant was calculated using the equation of [16]: RGR (mg/day) = (M2 − M1)/D; where M1 is the mass of the larvae at infestation, M2 is the mass of the larvae at the end of the experiment and D is the duration of the experiment (in days).

The same protocol was used for larval development on ear and silk. For each inbred line, a single larva and a whole shelled ear were placed in jars of 500 mL. The shelled ears were changed after 3 days. In addition, for each inbred line, approximately 150 mg of silk and a single neonate larva were placed in vials of 25 mL. Silks were changed every day. On shelled ear and silks, only RGR (the most significant parameter between lines for plant's leaves, see results) after 7 and 10 days of feeding, respectively, were assessed. All experiments were carried out at 25 ± 1 °C, RH of 75% ± 5 and 12 L:12 D. A total of 140, 75 and 20 larvae were evaluated per inbred line for leaf, silks and shelled ear, respectively.

### 2.6. Antibiosis Assessment under Net House Conditions

Antibiosis was assessed under net house conditions using leaves, silks and ears. The three experiments were conducted separately in different net houses. Using a camel brush, five neonate larvae, the minimum number of infestation to see differences between susceptible and resistant maize inbred lines (personal observations), were applied to the whorls of each plant. Open plastic sheaths filled with water were placed around each plot of Figure 1 to minimize larval movements between different plots or inbred lines. Two neonate larvae, the maximum number of infestation for silks and ears to avoid to be completely eaten after 14 days of infestation (personal observations), were used in the silk and ear experiments whereby plants in stage R1 (7 days after silking) and R3 (milking) [13] were used. In the ear experiment, the silk was removed. Silks or ears were covered with a small net bag upon infestation to minimize larval escape from these plants' organs.

The RGR of larvae after 7 and 14 days of feeding on leaves and after 14 days on both silks and ears were evaluated for each inbred line.

For each inbred line, a total of 40, 35 and 140 larvae were assessed for leaf, silks and ears, respectively.

*2.7. Antixenosis Assessment under Laboratory Conditions*

Antixenosis was evaluated in the laboratory on leaves, silk and grain. The experiment was performed under binary and multiple choice conditions in Petri dishes (9 cm). A Petri dish with wet filter paper was divided equally into two compartments for binary choice experiments following the protocol described by [17,18]. For multiple choice experiments, since there were seven inbred lines to be tested, a petri dish was divided into seven compartments to test the seven inbred lines together. A leaf portion of 3 cm$^2$, 100 milligrams of silk and a single grain was placed in each compartment of the petri dish for the leaf, silk and grain experiments, respectively. Ten neonate larvae were released in the center of each dish [19]. Each dish was covered with aluminum foil. After 1 h, 6 h and 16 h, the number of neonates found on each leaf portion, silk and grain were recorded. For the binary choice experiment, the feeding preference tests were first validated using non-preferred (neem) vs. preferred plant (maize) leaves for feeding.

The experiment was repeated 24 and 80 times for the binary and multiple choices, respectively.

*2.8. Data Analyses*

− For antibiosis assessments

The analysis of variance (ANOVA) was conducted using the generalized linear model (GLM) for data on insect development parameters in the antibiosis experiment. The GLM with the Gaussian distribution was used to analyze data on RGR, larval development time, weight of the male and female pupae, female pre-oviposition period and female oviposition period. The GLM with the Poisson distribution was used for the total number of eggs. The GLM with the Binomial distribution was used to analyze data on the number of eggs hatched and larval survival. Mean separation using Tukey's contrast test was performed for various parameters of the maize inbred lines using the multcomp package [20].

− For antixenosis assessments

To compare inbred lines for percentage of FAW neonates, data were analyzed using non-parametrical tests by the Wilcoxon rank sum test for the binary choice experiment and the Kruskal–Wallis rank sum test for the multiple choice experiment. The dunn.test package version 1.3.5 [21] was used for these analyses.

All statistical analyses were performed using R statistical software version 3.5.1 (R Core Team, 2018) [22].

**3. Results**

*3.1. Antibiosis Assessments*

There were significant differences in larval development on leaves of the different inbred lines (Table 1). The three resistant lines (Mp716, CML71 and CKSBL10008) showed a significantly longer larval development time compared to the susceptible lines CML444 and CKSBL10025. The weights of male pupae from the resistant lines Mp716, CML71, CML125 and CKSBL10008 were significantly lower. Larval survival, sex ratio and weight of female pupae did not vary significantly among the tested inbred lines.

**Table 1.** Larval development time (in days), larval survival (%), sex ratio and weight of male and female pupae (mg) from *Spodoptera frugiperda* larvae fed on leaf portions from different maize inbred lines under laboratory conditions. For each parameter, mean [1] ± SE are presented.

| Inbred Line | Larval Development Time n = 62–82 | Larval Survival n = 103–116 | Sex Ratio n = 11–15 | Weight of the Male Pupae n = 44–61 | Weight of the Female Pupae n = 38–61 |
|---|---|---|---|---|---|
| Mp716 | 17.7 ± 0.1 a | 82.7 ± 0.03 | 0.6 ± 0.08 | 197.2 ± 3.2 a | 188.3 ± 3.1 |
| CML71 | 17.7 ± 0.1 a | 73.3 ± 0.04 | 0.5 ± 0.08 | 191.6 ± 4.3 a | 186.4 ± 5.0 |
| CKSBL10008 | 17.9 ± 0.1 a | 80.3 ± 0.03 | 0.5 ± 0.04 | 197.1 ± 4.2 a | 181.4 ± 2.0 |
| CML125 | 17.0 ± 0.00 b | 73.4 ± 0.04 | 0.6 ± 0.09 | 191.8 ± 3.8 a | 187.0 ± 3.2 |
| CML370 | 17.1 ± 0.1 b | 82.7 ± 0.03 | 0.5 ± 0.01 | 201.9 ± 3.5 ab | 178.5 ± 2.9 |
| CKSBL10025 | 17.2 ± 0.1 b | 77.9 ± 0.04 | 0.6 ± 0.07 | 205.0 ± 2.8 ab | 188.8 ± 4.5 |
| CML444 | 17.0 ± 0.1 b | 81.3 ± 0.03 | 0.5 ± 0.06 | 214.9 ± 3.4 b | 193.3 ± 5.4 |
| *p*-Value of the ANOVA | ≤0.0001 | 0.2327 | 0.76 | ≤0.0001 | 0.1254 |

[1] Means within each column followed by different letters are significantly different at 5% probability level, according to Tukey's HSD test.

The pre-oviposition period varied significantly among the inbred lines (Table 2). A longer pre-oviposition period was found on Mp716 and CML71 as compared to the females coming from the susceptible lines CML444 and CKSBL10025. Female fecundity (reflected to the number of eggs laid per female), fertility (reflected by % of eggs that hatched) and the oviposition period did not vary significantly between inbred lines.

**Table 2.** Females' pre-oviposition period (in days) after mating, oviposition period (in days), number of eggs laid per female and percentage of eggs hatched from larvae of *Spodoptera frugiperda* fed on leaf portions of different maize inbred lines under laboratory conditions. For each parameter, mean [1] ± SE are presented.

| Inbred Line | Pre-Oviposition Period n = 13–23 | Oviposition Period n = 15–23 | Number of Eggs Laid per Female n = 14–21 | % Eggs Hatched n = 15–22 |
|---|---|---|---|---|
| Mp716 | 2.4 ± 0.1 a | 5.7 ± 0.6 | 589.6 ± 41.2 | 95.8 ± 2.2 |
| CML71 | 2.5 ± 0.2 a | 5.2 ± 0.9 | 505.1 ± 68.4 | 98.3 ± 0.9 |
| CKSBL10008 | 2.3 ± 0.1 ab | 4.3 ± 0.6 | 484.9 ± 67.4 | 95.3 ± 2.3 |
| CML125 | 2.3 ± 0.1 ab | 5.0 ± 0.7 | 420.9 ± 84.7 | 98.9 ± 0.6 |
| CML370 | 2.0 ± 0.00 b | 3.1 ± 0.5 | 411.1 ± 62.9 | 92.1 ± 4.4 |
| CKSBL10025 | 2.0 ± 0.00 b | 5.0 ± 0.6 | 545.0 ± 47.4 | 95.7 ± 2.1 |
| CML444 | 2.0 ± 0.00 b | 4.9 ± 0.7 | 528.8 ± 59.5 | 92.6 ± 2.3 |
| *p*-Value of ANOVA | 0.0145 | 0.1539 | 0.3386 | 0.32 |

[1] Means within each column followed by different letters are significantly different at 5% probability level according to Tukey's HSD test.

Table 3 presents the RGR of larvae fed on leaves, silks and ears under both laboratory and net house conditions. On leaves, the larvae exhibited a significant lower RGR on the resistant inbred lines CML71 and CKSBL10008 after 10 days under the laboratory conditions, and 7 days and 14 days under the net house conditions as compared to susceptible lines CML444 and CKSBL10025. The larvae that fed on Mp716 also showed a significantly lower RGR as compared to CML444 under laboratory conditions. Under laboratory conditions, larvae had a highest RGR on silks of the resistant line CKSBL10008 than on silks of the other inbred lines. No difference was found on larvae feeding on silks under net house conditions. On shelled ears under laboratory conditions, the highest RGR was obtained in the susceptible line CKSBL10025.

**Table 3.** Relative growth rates (RGR, mg/d) of *Spodoptera frugiperda* larvae after 7, 10 and 14 days of infestation on maize inbred lines. For each parameter, mean [1] ± SE are presented.

| | RGR under Lab Conditions | | | RGR under Net House Conditions | | | |
|---|---|---|---|---|---|---|---|
| **Inbred Line** | **On Leaves (10 Days) n = 116–127** | **On Silks (10 Days) n = 40–74** | **On Shelled Ear (7 Days) n = 10–17** | **On Leaves (7 Days) n = 24–32** | **On Leaves (14 Days) n = 23–31** | **On Silks (14 Days) n = 10–31** | **On Ears (14 Days) n = 46–136** |
| Mp716 | 10.6 ± 0.3 bc | 2.8 ± 0.1 a | 3.4 ± 0.4 a | 1.2 ± 0.1 ab | 16.4 ± 1.0 abc | 1.5 ± 0.3 | 11.3 ± 0.8 ab |
| CML71 | 8.1 ± 0.4 a | 2.7 ± 0.1 a | 3.2 ± 0.4 a | 1.2 ± 0.1 ab | 15.0 ± 1.4 ab | 2.3 ± 0.5 | 10.3 ± 0.5 a |
| CKSBL10008 | 9.2 ± 0.4 ab | 4.6 ± 0.2 b | 3.8 ± 0.4 a | 1.0 ± 0.1 a | 13.8 ± 0.6 a | 3.5 ± 0.8 | 17.7 ± 0.8 c |
| CML125 | 12.0 ± 0.5 cd | 2.3 ± 0.1 a | 4.5 ± 0.5 a | 1.7 ± 0.1 b | 18.1 ± 1.2 bc | 3.2 ± 0.7 | 10.3 ± 0.6 a |
| CML370 | 11.3 ± 0.4 cd | 2.7 ± 0.1 a | 3.2 ± 0.6 a | 1.4 ± 0.1 ab | 19.1 ± 1.2 c | 2.5 ± 0.5 | 13.5 ± 0.9 b |
| CKSBL10025 | 11.2 ± 0.4 cd | 2.7 ± 0.1 a | 9.0 ± 1.2 b | 1.7 ± 0.1 b | 19.2 ± 1.2 c | 2.5 ± 0.5 | 20.1 ± 1.7 c |
| CML444 | 13.0 ± 0.5 d | 2.4 ± 0.1 a | 4.9 ± 0.8 a | 3.6 ± 0.3 c | 28.1 ± 1.0 d | 2.7 ± 0.44 | 11.4 ± 0.7 ab |
| *p*-Value of ANOVA | ≤0.0001 | ≤0.0001 | ≤0.0001 | ≤0.0001 | ≤0.0001 | 0.1188 | ≤0.0001 |

[1] Means within each column followed by different letters are significantly different at 5% probability level according to Tukey's HSD test.

### 3.2. Antixenosis Assessments on Leaves

#### 3.2.1. Binary Choice Tests

The feeding preference tests were validated using a non-preferred plant for feeding (neem leaves) vs. a preferred plant for feeding (maize leaves). After 1 h of bioassay, 70% of the larvae preferred to feed on maize leaves followed by 90% after 6 h and 100% after 16 h respectively (Figure 2).

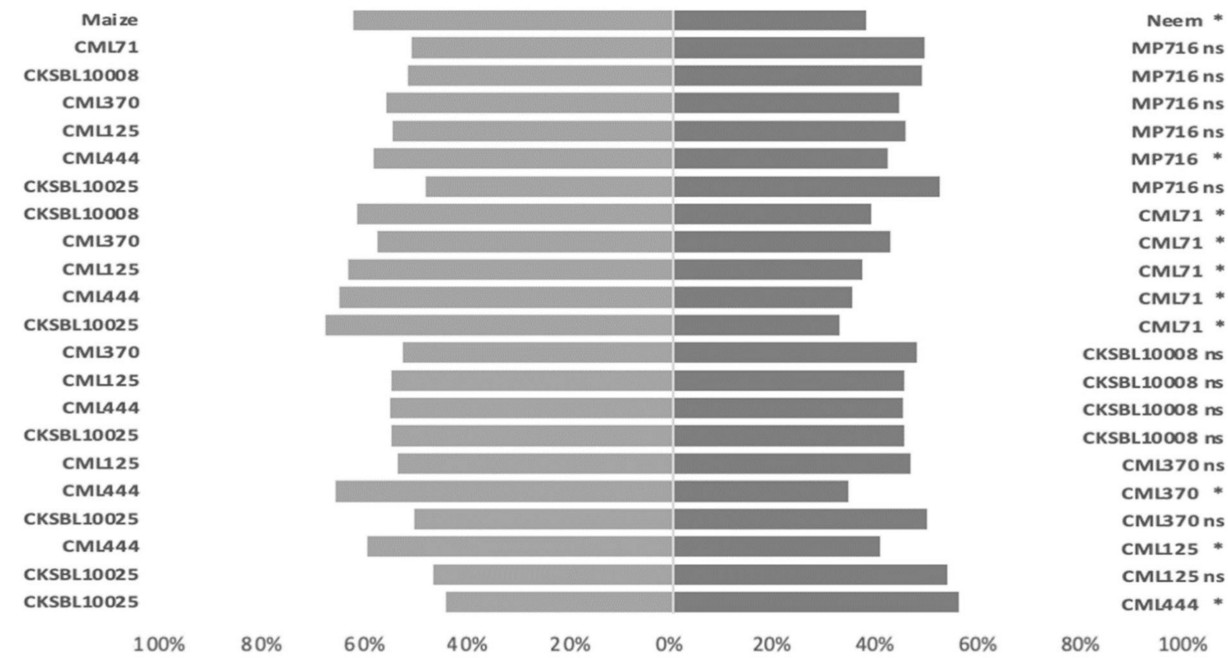

**Figure 2.** *Cont.*

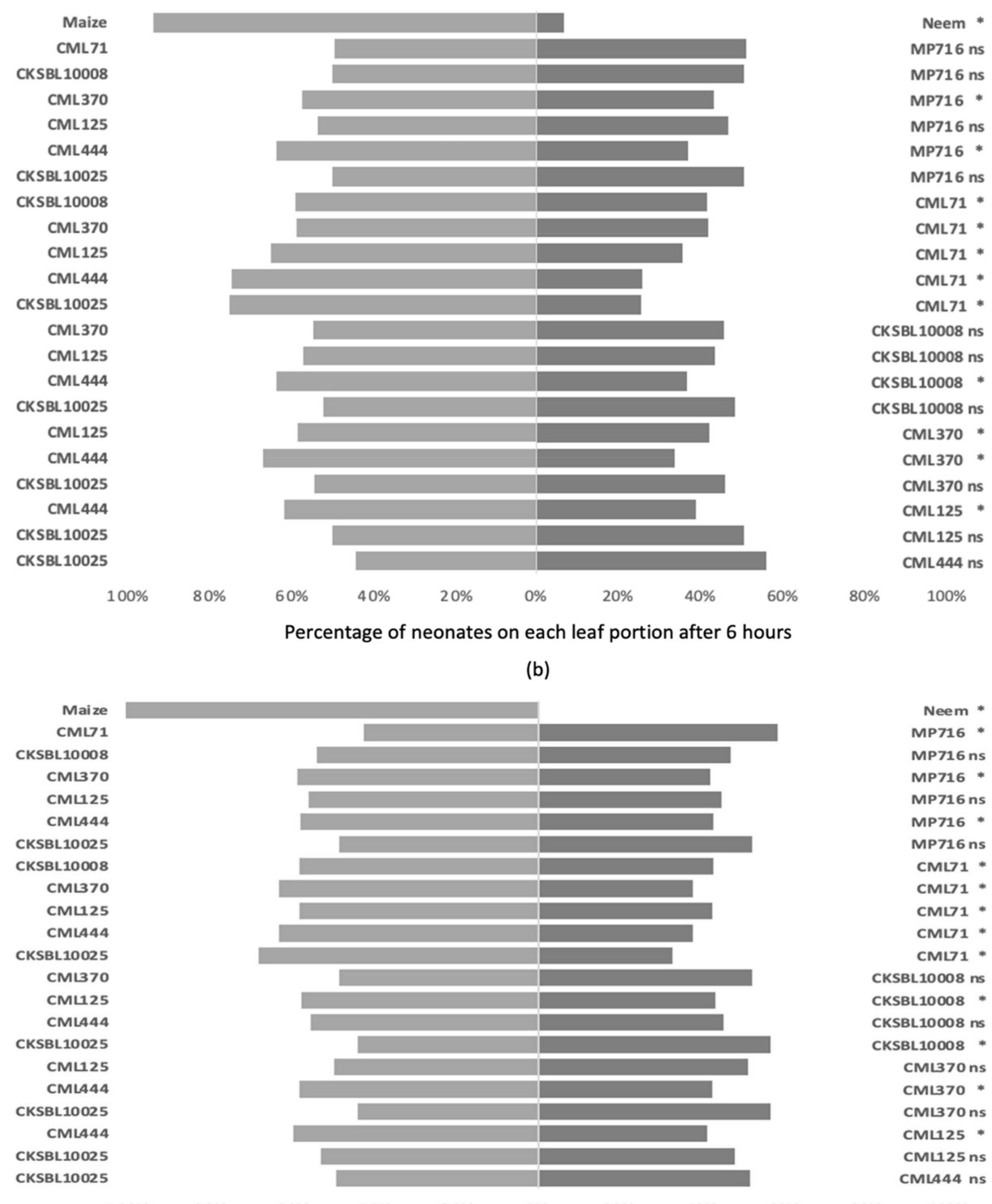

**Figure 2.** Mean percentage of *Spodoptera frugiperda* neonates found on leaf portions of each inbred line under binary choice conditions between inbred lines (n = 21) in a Petri dish after 1 (**a**), 6 (**b**) and 16 h (**c**) of bioassay. Asterisks (*) represent significant differences and (ns) represent no significant differences at 5% level (Wilcoxon rank sum test).

Among the inbred lines tested, CML71 was less preferred after 1, 6 and 16 h of bioassay than the other inbred lines respectively (Figure 2). Mp716, CML370 and CKSBL10008 were less preferred than CML444 after 1 and 6 h. At 1 and 6 h, there was no difference between CML71 and Mp716. There was no significant difference observed between CSBL10008 and Mp716 after exposition for 1, 6 and 16 h.

### 3.2.2. Multiple Choice Tests

In the multiple choice tests, significant differences were also found among the inbred lines (Figure 3). The resistant CML71 was less preferred after 1, 6 and 16 h of bioassay than the other inbred lines. Mp716 and CML125 were less preferred after 1 and 6 h than the susceptible CML444. CKSBL10008 was also less preferred than CML444 after 1, 6 and 16 h. However, CML370 did not show a lower preference than CML444 as found in the binary choice test and was even the most preferred after 16 h. CKSBL10008 was less preferred than CML444 after 1, 6 and 16 h.

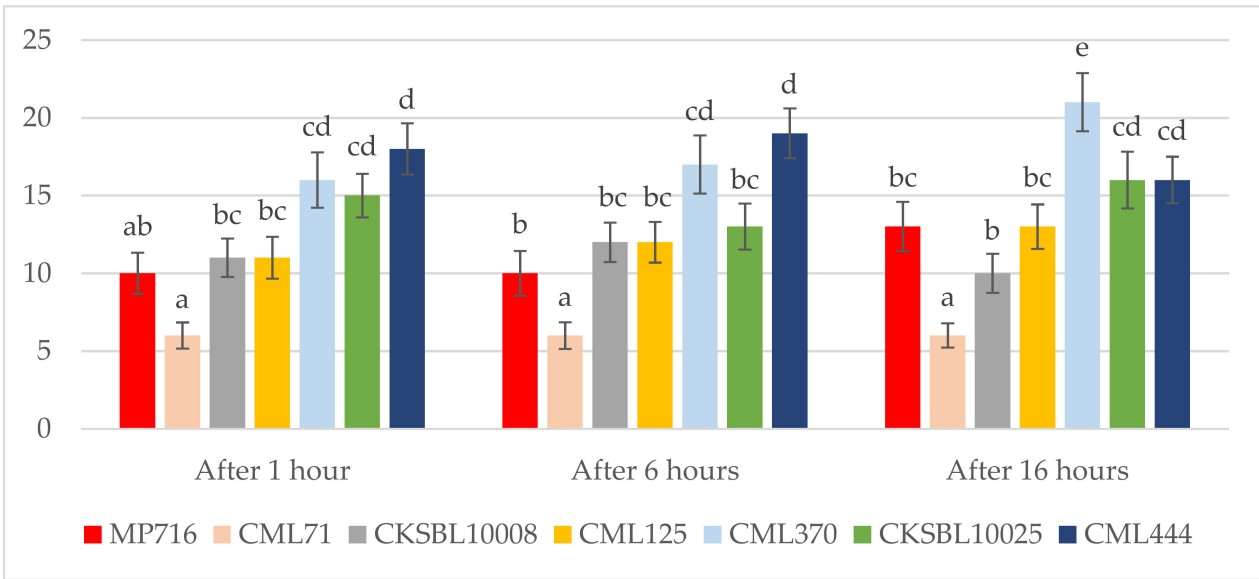

**Figure 3.** Mean percentage (±SE) of *Spodoptera frugiperda* neonates found on leaf portions of each inbred line under multiple choice conditions after 1, 6 and 16 h of bioassay among inbred lines (n = 71–75) in a Petri dish. Means within each experimental time followed by different letters are significantly different at 5% probability level according to Dunn's Test.

### *3.3. Antixenosis Assessments on Maize Silks and Grains*

### 3.3.1. Binary Choice Tests

There were significant differences between the inbred lines for feeding preference on silks (Figure 4) and grains (Figure 5). On silks, Mp716 and CML125 were less preferred than CML444 after 6 h and 16 h. On leaves, CKSBL10008 was less preferred than CML444 after 6 h. On grains, only CKSBL10008 was less preferred after 6 h as compared to CML444.

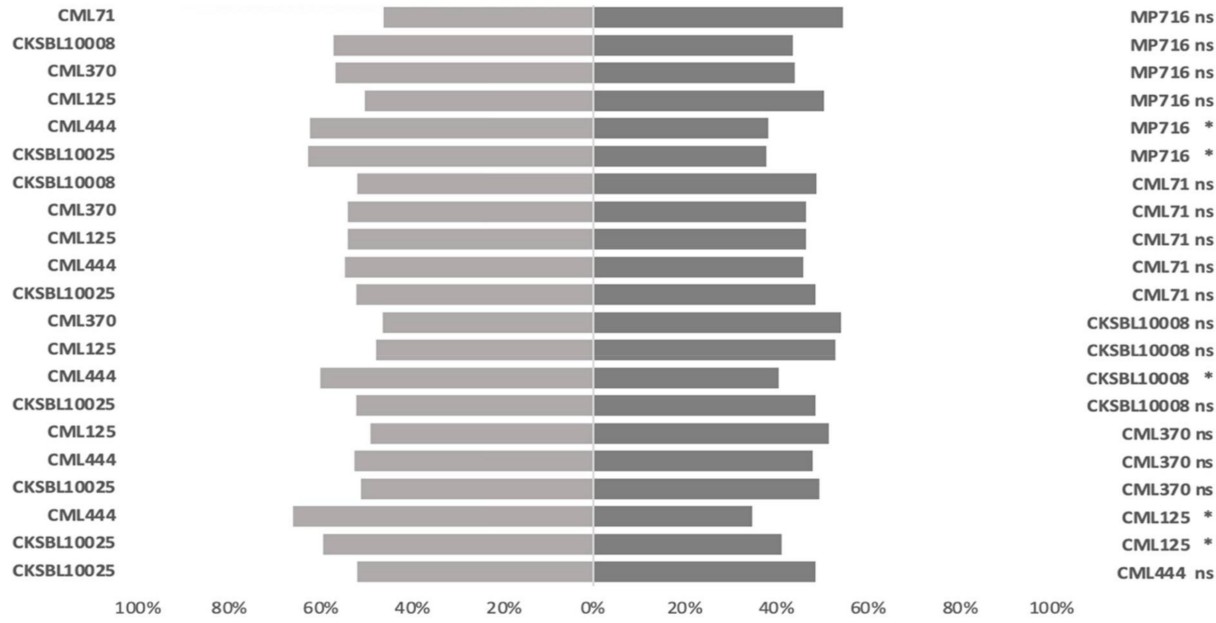

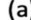

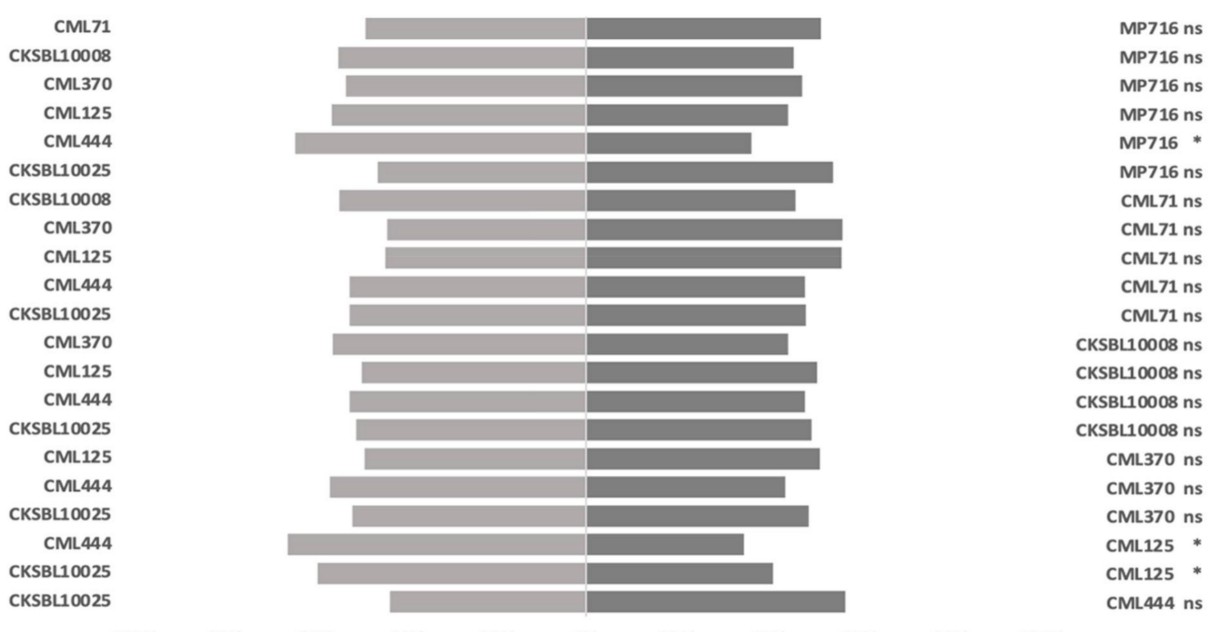

**Figure 4.** Mean percentage of *Spodoptera frugiperda* neonates found on silk of each inbred line under binary choice conditions after 6 (**a**) and 16 h (**b**) (n = 24) of bioassay between maize inbred lines in a Petri dish. Asterisks (*) represent significant differences and (ns) represent no significant differences at 5% level (Wilcoxon rank sum test).

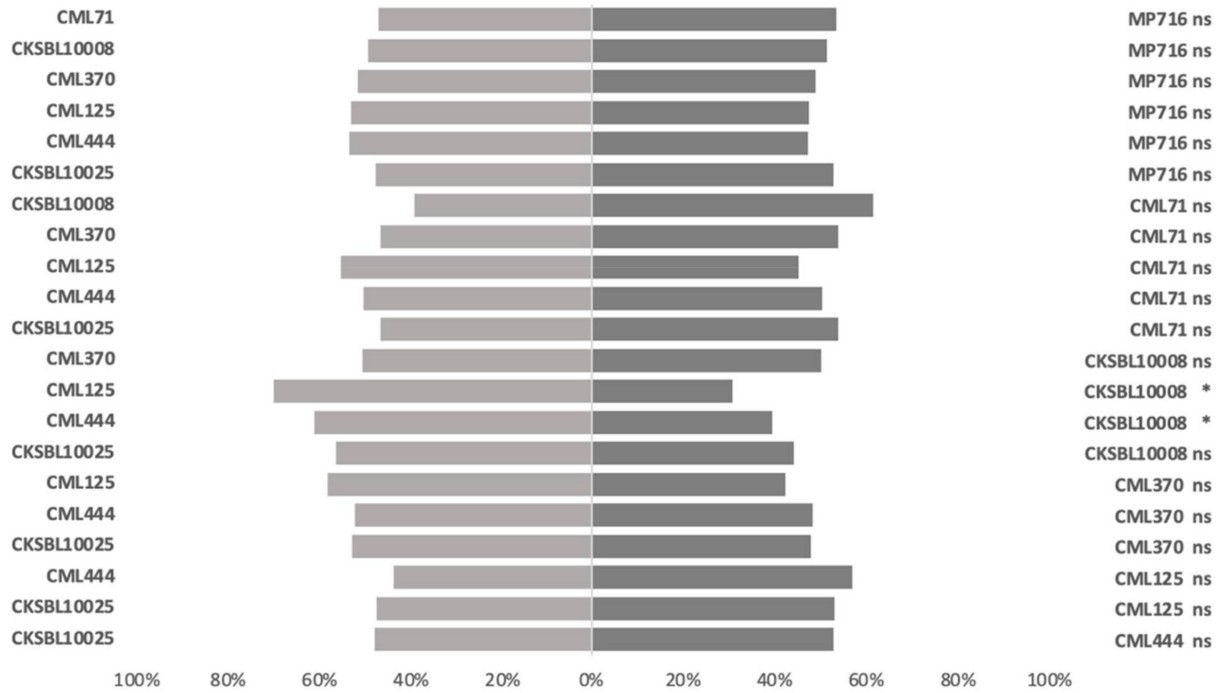

Percentage of neonates on each grain after 6 hours

(a)

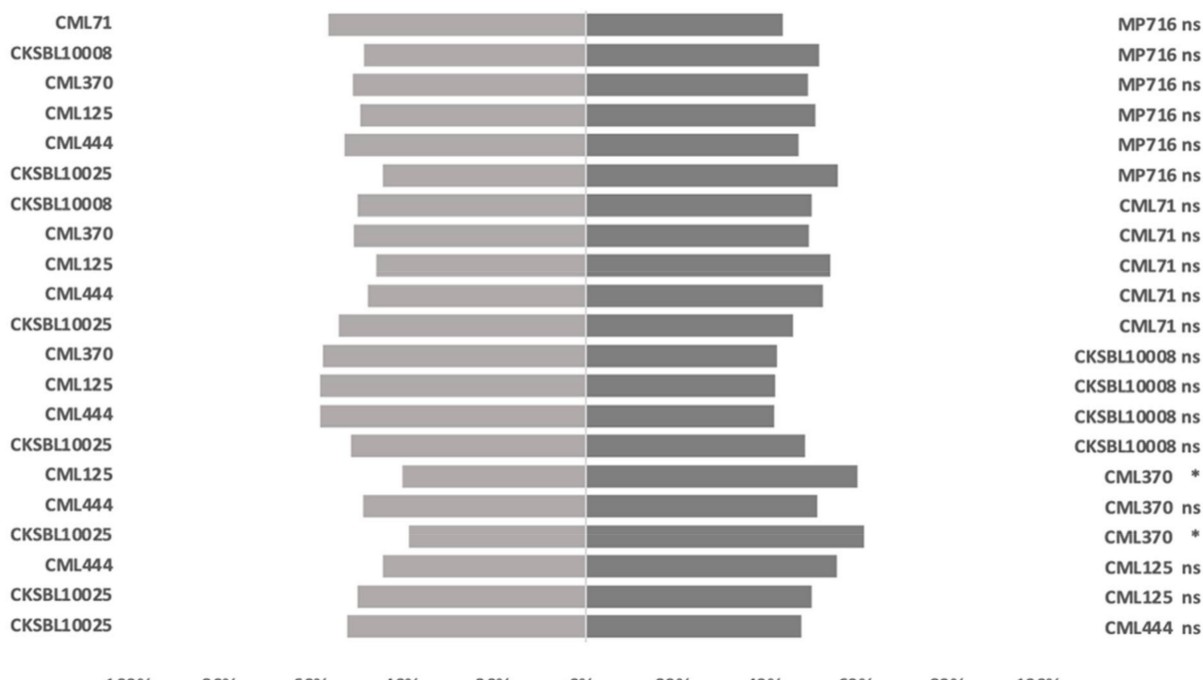

Percentage of neonates on each grain after 16 hours

(b)

**Figure 5.** Mean percentage of *Spodoptera frugiperda* neonates found on grain of each inbred line under binary choice conditions after 6 (**a**) (n = 20) and 16 (**b**) (n = 24) hours of bioassay between maize inbred lines in a Petri dish. Asterisks (*) represent significant differences and (ns) represent no significant differences at 5% probability level (Wilcoxon rank sum test).

### 3.3.2. Multiple Choice Tests

On silks, a significant difference of feeding preference was observed (Figure 6). The inbred lines Mp716 and CML71 were less preferred than CKSBL10008 after 6 h. The resistant inbred lines Mp716, CML71, CKSBL10008, CML125 and CML370 were less preferred than the susceptible inbred line CML444 after 16 h of bioassay.

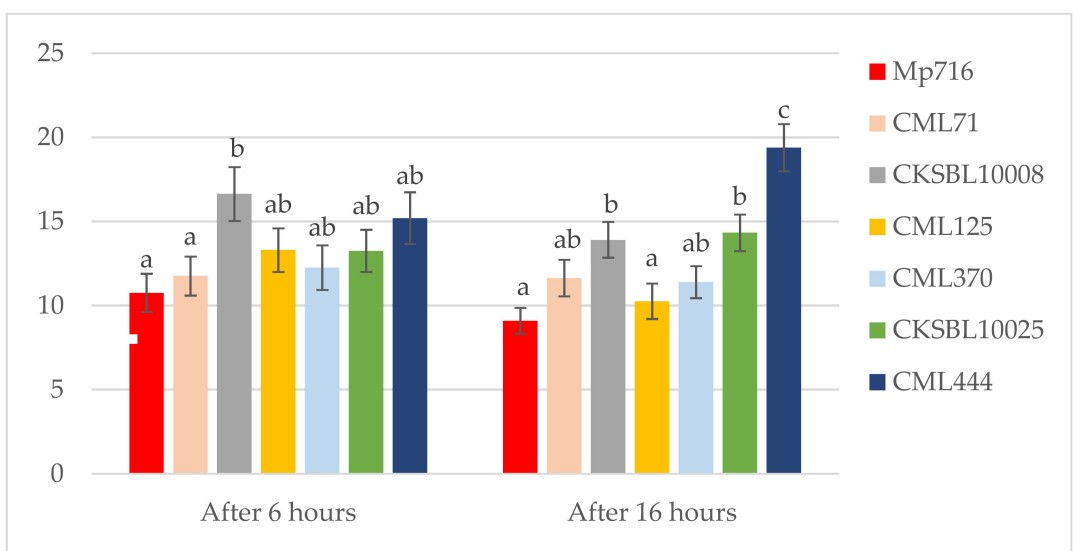

**Figure 6.** Mean percentage (±SE) of *Spodoptera frugiperda* neonates found on silk portions of each inbred line under multiple choice conditions after 6 (n = 80) and 16 (n = 78) hours of bioassay in a Petri dish. Means within experimental time followed by different letters are significantly different at 5% probability level according to Dunn's Test.

Feeding preference assessed on maize kernels did not reveal significant differences among inbred lines in multiple choice tests (Figure 7) after 6 and 16 h. Overall, the percentage by inbred lines was between 10 and 15% after 6 or 16 h of exposure.

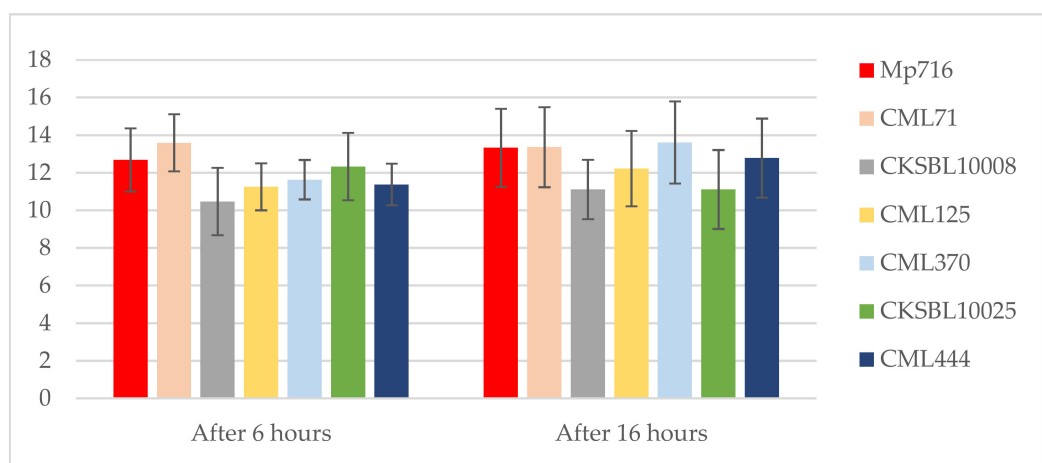

**Figure 7.** Mean percentage (±SE) of *Spodoptera frugiperda* neonates found on grain of each inbred line under multiple choice conditions after 6 (n = 41) and 16 (n = 36) hours of bioassay in a Petri dish.

### 4. Discussion

Larvae fed with leaves from the resistant inbred lines Mp716, CML71 and CKSBL10008 weighed significantly less and took longer to pupate than those fed with leaves from the susceptible inbred lines under both laboratory and greenhouse conditions. Among the maize inbred lines tested, CML71 and CKSBL10008 showed the same level of antibiosis as

the "resistant" control, Mp716. A long development time and low weight of FAW larvae was reported by [23] and [24] for an FAW-resistant inbred line (Mp708), which is a parent to Mp716 [9]. Wiseman et al. [8] also reported a high level of antibiosis in population MpSWCB-4, which is derived from population Antigua Gpo2 [25] the source germplasm for inbred line CML71 [26]. Larval development times of the three resistant inbred lines in our study (Mp716, CML71 and CKSBL10008) are slightly higher than that of the resistant Mp708 and like that of the resistant FAW7050 maize inbred reported by [24] in the United States, but higher than those obtained by [19] on maize cultivars used in Kenya. The results of the net house experiment showed that RGRs of larvae fed on three resistant inbred lines were approximately two to three times lower than that of the susceptible inbred line, CML444. These decreases are similar to those reported by [24] for larvae fed for one week on the resistant lines Mp708 and FAW7050 and the susceptible genotype Ab24E. Although, reported as "resistant" to FAW in field tests [6], the inbred lines CML125 and CML370 did not show a reduced larval development in the laboratory as compared to the susceptible inbred lines CKSBL10025 and CML444.

Under laboratory conditions, the lowest RGR was obtained on CML71 whereas it was obtained on CKSBL10008 under net house conditions. This can be because the larvae fed on portions of leaves in the laboratory while in the net house they were feeding on whole plants. Plants fight herbivores by morphological, biochemical, and molecular mechanisms. This fight involves constitutive and induced defenses (see [27] for review). In maize, [28] found that FAW feeding induced foliar RIP2 protein accumulation, a protein which retarded larval growth considerably. Similarly, [29] showed that resistant varieties of maize can produce a resistance factor(s) to inhibit the chymotrypsin activity of *Ostrinia furnacalis* (Lepidoptera: Crambidae) and suppress larval growth. Shivaji et al. [30] also showed that jasmonic acid, a compound that plays an important role in the defense of maize against FAW, is produced constitutively in certain genotypes, suggesting that they are "primed" to respond rapidly to an attack and this is visible when the plants are entire but not in pieces. In this context, although antibiosis resistance of CML71 and CKSBL10008 found under laboratory conditions was confirmed under net house conditions, CKSBL10008 would have had a better induced defense capacity on whole plants than on leaf portions, which could explain the difference in results obtained between the laboratory and the net house.

Among all the parameters evaluated in this study, only RGR and the larval development time were found to be the most relevant to assess antibiosis resistance but surprisingly not the larval survival, confirming results by [31] who reported significant effects on larval weights and on the duration of larval development but not on larval survival in their study using other maize genotypes.

Maize silk has been reported to resist FAW larvae because of their maysin concentration [32]. This could not be verified in our study. Similarly, silk of lines (Dixie 18 and 471-U6 X 81) resistant to *Helicoverpa zea* (Lepidoptera, Noctuidae) was not shown to affect weight and survival of larvae [33,34]. The resistance of the kernels of these resistant genotypes was explained by the presence of tight husks, long silk channels and large amounts of silks that maintained a high moisture content [35]. In fact, [36] later found little or no maysin in silks of these cultivars. This may explain why in our study, inbred lines that showed resistance in leaves did not show this resistance in silks and grains. Interestingly, the most resistant genotypes (CML71) were the least preferred for feeding on leaves (i.e., antixenosis resistance) in both binary choice and multiple choice tests, after 1, 6 and 16 h of bioassay. CML71 is considerably less preferred by FAW neonates compared to the susceptible inbred lines CML444 and CKSBL10025, which is similar to the feeding preference results obtained by [37] between the resistant and susceptible maize hybrids. Likewise, [8] found a non-preference for feeding in leaves of the resistant genotypes Antigua 2D-118 and MpSWCB-4. Yang et al. [38] found that cuticular lipid was involved in the non-preference of MpSWCB-4; this genotype has a similar genetic background as CML71 as both are derived from Antigua Gpo2 [25,26]. Similarly, a low feeding preference of FAW larvae was also observed in fresh leaf portions of a resistant inbred line Mp708 compared to a susceptible inbred

line Tx601 [39]. These authors identified (E)-β-caryophyllene as the volatile compound involved in this non-preference.

Wiseman et al. [40] found that silks of a resistant genotype Zapalote Chico were less preferred by the larvae *H. zea* compared to those of susceptible genotype Stowell's Evergreen. In our study, the silks of the resistant inbred lines Mp716 and CML125 were the least preferred, followed by CKSBL10008. As for the antibiosis, inbred lines that were less preferred by neonates on leaves did not show non-preference on silks and grains, suggesting that the resistance factors (chemical or physical) linked to antixenosis are most probably located in leaves.

As highlighted by [41], our study also showed that FAW resistance is acting through antixenosis and antibiosis mechanisms in germplasm lines. Among the inbred lines studied here, CML71 is revealed as a highly promising line for use in breeding for native genetic resistance to FAW in tropical maize. Its resistance to FAW can be due to chemical characteristics present in the leaves as shown by [42] from a specific maize race to red spider mite. Chemical characteristics have been reported to confer resistance to FAW damage in maize such as the presence of jasmonic acid [30], (E)-β-caryophyllene [39], cuticular lipids [38,43,44], silica [45,46], high induced defensive transcriptomic signatures and higher levels of benzoxazinoids [47].

## 5. Conclusions and Perspectives

In conclusion, the inbred line CML71 showed good antibiosis and good non-preference for feeding (or antixenosis) as a mechanism of resistance to FAW larvae on leaves, but it did not show resistance on silks and grains. CKSBL10008 also showed good antibiosis on leaves and some non-preference for feeding on leaves, silks and ears. Ortega et al. [48] mentioned that knowledge of the mechanisms involved in plant resistance can help in the selection of genotypes with this characteristic, in order to increase the efficiency of the breeding program. The information on the resistance mechanism of these inbred lines is relevant to the program initiated by CIMMYT-Kenya to develop FAW-resistant tropical maize hybrids, and local and exotic lines can contribute on that [49]. Moreover, our study shows that it is important to complement the screening of plant resistance based on the assessment of damage and injury caused by the insect pest with a more accurate assessment, as was carried out in this study. However, it is not necessary to study the effect of a plant's genotype on all biological parameters of the insect to detect resistance, only the assessment of larval RGR after 10 days and the feeding preference after 16 h appears to be sufficient under laboratory conditions. This is important to know in a context to find out low time-consuming and low-cost assays to identify plants potentially carrying resistance traits, within a high number of traditionally bred varieties or material derived from global germplasm [50].

In addition, the reduced development and non-preference for feeding of larvae on leaves of the resistant inbred line CML71 in our study suggest the involvement of chemical characteristics present in the leaves. Future studies are needed to identify the chemical compounds involved and to study their mode of action on the feeding behavior and development of FAW larvae.

**Author Contributions:** Conceptualization, O.N.-Y., A.Y.B., G.O.O., C.M., D.M., Y.B., B.M.P., F.M.-P. and P.-A.C.; methodology, O.N.-Y., A.Y.B., G.O.O., C.M., D.M., Y.B., N.-M.M., B.M.P. and P.-A.C.; formal analysis, O.N.-Y. and N.-M.M.; resources, A.Y.B.; writing—original draft preparation, O.N.-Y., A.Y.B. and P.-A.C.; writing—review and editing, O.N.-Y., A.Y.B., F.M.-P. and P.-A.C.; supervision, A.Y.B., F.M.-P. and P.-A.C.; project administration, A.Y.B., F.M.-P. and P.-A.C.; funding acquisition, N.-M.M., A.Y.B., B.M.P. and P.-A.C. All authors have read and agreed to the published version of the manuscript.

**Funding:** This research was funded by the Université Kongo (U.K.)-DRC through a PhD fellowship under the grant number 006/U.K./REC/OGN/2019, by the 'Institutes de Recherche pour le Développement' (IRD)-France through the IRD Collaborative Research project (grant number B4405B) and the integrated pest management strategy to counter the threat of invasive fall armyworm to food

security in eastern Africa (FAW-IPM) (grant number DCI-FOOD/2017/) financed through the European Union. This study was undertaken through the financial support from the United States Agency for International Development (USAID) Bureau for Resilience and Food Security to CIMMYT under Grant # BFS-IO-17-00005 as part of Feed the Future activity, under the Fall Armyworm Management Project to CIMMYT, and the CGIAR Research Program on Maize (MAIZE) Windows 1 and 2.

**Data Availability Statement:** Not applicable.

**Acknowledgments:** The authors wish to thank *icipe* Capacity Building Program (DRIP), CIMMYT-Nairobi and KALRO-Katumani for hosting the PhD student. The authors are grateful for the collaboration extended by the Kenya Agricultural and Livestock Research Organization (KALRO) Research Centers at Machakos and Kiboko. The authors would also like to thank Fritz Schulthess for the revision of the manuscript.

**Conflicts of Interest:** The authors declare no conflict of interest. The funders had no role in the design of the study; in the collection, analyses, or interpretation of data; in the writing of the manuscript; or in the decision to publish the results.

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
