# Peer review of "Assessment of Resistance Mechanisms to Fall Armyworm, Spodoptera frugiperda in Tropical Maize Inbred Lines"

_agronomy, doi:10.3390/agronomy13010203_

Round 1

Reviewer 1 Report

The manuscript “Assessment of resistance mechanisms to Fall Armyworm,  Spodoptera frugiperda in tropical maize inbred lines.” found among the seven maize inbred lines tested, two, namely CML71 and CKSBL10008, exhibited the highest level of antibiosis resistance. CML71 showed also a good non-preference compared to other tested lines. The non-preference for feeding, lower relative growth rate and longer developmental time of larvae on CML71 suggest a biochem- ical involvement resistance to FAW. However, there are some problems to be addressed in the current version before it can be accepted for publication.

1. Why did the author choose these seven species for experiments? Since four of them were known to be resistant to FAW and the other two were sensitive. Why not experiment with other new varieties? In addition, Mp716 was selected as the resistant control, and whether the sensitive variety was selected as the negative control?

2. Line 88-91. The colonies were maintained at 25 ± 1°C, RH of 75% ± 5 and a photoperiod of 12:12 (L:D) h.

Line 103-107. The net house experiment trial was carried out at 28 ℃, 48%RH, 12L: 12D. 

Reducing the relative humidity from 75% to 48% does this affect the mortality of newborn larvae?

3. Line 138 and 141. How do neonate larvae and pupae of FAW distinguish between male and female?

4. Line 150. A total of 140, 75 and 20 larvae were evaluated per inbred line for leaf, silks and shelled ear, respectively .

Line 163. For each inbred line, a total of 40, 35 and 140 larvae were assessed for leaf, silks, and 163 ears respectively .

How to explain this difference?

5. Line 230-232. Under laboratory conditions, larvae had a highest RGR on silks of the resistant line CKSBL10008 than on silks of the other inbred lines. No difference was found on larvae feeding on silks under net house conditions. How can you explain this conflict?

Author Response

  1. Why did the author choose these seven species for experiments? Since four of them were known to be resistant to FAW and the other two were sensitive. Why not experiment with other new varieties? In addition, Mp716 was selected as the resistant control, and whether the sensitive variety was selected as the negative control?

Reply: As mentioned now in the text an intensive screening against FAW was done by artificial infestation in greenhouse conditions in Kenya between 2017 and 2018 of about 3000 inbred lines available in the germplasm collection of CIMMYT. Among these 3000 lines, only 4 showed to be resistant to FAW, namely CML71, CML125, CML370 and CKSBL10008, justifying why these lines were chosen and not others. To select sensitive controls we used two lines, namely CML444 and CKSBL10025 among the 3000 lines screened, which were the most sensitive lines found in the extensive screening of CIMMYT.

  1. Line 88-91. The colonies were maintained at 25 ± 1°C, RH of 75% ± 5 and a photoperiod of 12:12 (L:D) h. Line 103-107. The net house experiment trial was carried out at 28 ℃, 48%RH, 12L: 12D. Reducing the relative humidity from 75% to 48% does this affect the mortality of newborn larvae?

Reply: As described at the sub-section 2.6, Antibiosis assessment under net house conditions, we did not evaluate the mortality of newborn larvae in the net house experiments since as a reduction the relative humidity from 75% to 48% might affect the mortality of newborn larvae. Also, the RGR was evaluated only on live larvae.

  1. Line 138 and 141. How do neonate larvae and pupae of FAW distinguish between male and female?

Reply: As described at the sub-section 2.5. Antibiosis assessment under laboratory conditions, the sex ratio of the progeny of each colony reared from each inbred line tested, approximately 200 neonate larvae from the hatched eggs of each female from each inbred line were reared on an artificial diet until pupation (the first stage to determine the sex of each individual). It is very easy and common to distinguish the sex of FAW pupae and moth pupae in general.

  1. Line 150. A total of 140, 75 and 20 larvae were evaluated per inbred line for leaf, silks and shelled ear, respectively.

Line 163. For each inbred line, a total of 40, 35 and 140 larvae were assessed for leaf, silks, and 163 ears respectively.

How to explain this difference?

Reply: This difference is explained by the availability of the infested plant between laboratory and net house conditions. More plants can be planted, but before the infestation some of them are dead or not growing well explaining these differences in replicates.

  1. Line 230-232. Under laboratory conditions, larvae had a highest RGR on silks of the resistant line CKSBL10008 than on silks of the other inbred lines. No difference was found on larvae feeding on silks under net house conditions. How can you explain this conflict?

Reply: This conflict can be explained by the different plant growing conditions between experiments realized under net house and laboratory conditions. For net house conditions the plants are planted directly into the soil whereas for laboratory conditions the plants are planted into pots. This can give some differences in silks of plants grown on pots but not or less differences in silks of plants grown directly on soil.

Reviewer 2 Report

Comments and Suggestions for Authors

Manuscript ID: agronomy-2129951

The paper entitled “of resistance mechanisms to Fall Armyworm, Spodoptera frugiperda in tropical maize inbred lines.” was carefully reviewed. The purpose of this work was to determine the antibiosis and antixenosis to FAW in selected maize inbred lines under laboratory and net house conditions.

The manuscript needs to be revised before considering publication in “Agronomy”.

Detailed comments:

Abstract

-          Line 20: “first reported in West Africa in 2016”. Please correct the sentence as indicated in the introduction section (Line 45; “…first reported in West and Central Africa in 2016 [2]”).

-          Add quantitative results to this section.

-          Improve on the conclusions of the abstract.

Introduction

-          Lines 51-53: “Most farmers in African countries are resource-constrained smallholders and usually face problems in effectively controlling insect pests like stem borers and FAW using insecticides”. Add a reference for this statement.

-          Lines 59-60: “several maize inbred lines resistant to S. frugiperda have been developed namely Mp708, Mp78:518 [6,7] and Mp716 [8]”. Several or three inbred lines? Please correct.

Materials and methods

-          Line 85: Stick to either “FAW” or “S. frugiperda” through the text.

-          “In addition, ears at the R1 silking stage (7 days after silking) and at R3 milk stage were used”. What were the ears used for? Please complete the sentence.

-          Line 106: delete “Kenya Agriculture and Livestock Research Organization (KALRO)” and only keep “KALRO” to avoid the redundancy in the text.

-          Lines 107-109: “Before planting, Emamectin benzoate (19 g/l) insecticide was sprayed on the ground and walls of the net house against insects that might interact with the experiment”. Indicate when the pesticide will be applied.

-          Line 111: “10 g of phosphate (DAP) were…”. Delete “phosphate” and keep “DAP” to avoid the repetition in the text.

-          Lines 132-133: “To estimate the sex ratio, approximately 200 neonate larvae from the hatched eggs of each female from each inbred line were reared”. Why is it around 200?  Please explain.

-          Lines 141-143: “RGRs were calculated by subtracting the average weight per larva on each inbred line at infestation from the average weight per larva on the same inbred line at the day of weighing, divided by the number of days in between”. This lengthy sentence should be replaced with a formula, in my opinion. Also, include the formulae for survival%, egg hatch%, adult emergence%, and so on.  

-          Lines 148-149: “On shelled ear and silks, only RGR after 7 and 10 days of feeding respectively were assessed”. Explain why only The RGR was evaluated. How about the other tested parameters?

-          Line 150: 2 Replace “(Light: Dark)” by (L: D).

-          Line 152: “2.6. Antibiosis assessment under net house conditions”. This section is unclear and should be rewritten in terms of experimental design and trial establishment (number of tested plants, plots, and replicates; ears and silk experiments?).

-          Line 155-158: Explain why each plant received 5 neonate larvae while the silk and ear experiments received only 2 neonate larvae.

-          Line 156: Indicate the properties of the plastic sheath used in the experiment.

-          Line 160: Indicate the properties of the net bag used in the experiment.

-          Line 165: “2.7. Antixenosis assessment under laboratory conditions”. Why not conduct the feeding preference tests in the field (e.g., in cages)? These tests will be used to validate the laboratory results.

-          Line 173: “Ten neonate larvae were released in the center of each dish”. FAW neonate larvae are cannibalistic, as we all know. Did you study the intraspecific competition of FAW larvae in the laboratory? Before performing the multiple-choice experiments, this parameter should be considered.

-          Line 179: Explain why the experiment was repeated 24 and 80 times for the binary and multiple-choice experiments, respectively.

-          I have some reservations about the statistical analyses. This section should be rewritten and divided into sub-titles to make each test easier to understand.

-          Lines 192-193: “the Wilcoxon rank sum test and Kruskal-Wallis rank sum test using dunn.test package version 1.3.5”. Please rewrite the sentence to avoid any confusion.

-          Line 194: “Statistical significance was defined at the 5% level for all comparisons”. Delete this sentence.

Results:

-          Table 1: "(number of females/total number)" should be removed from the title.

-          Line 224: Replace “the relative growth rates (RGR)” by “RGR”.

-          Line 246: Replace “line” by “lines”.

-          Lines 266-268: “Mp716 was less preferred than CML444 after 1, 6 and 16 hours. CML370 was also less preferred than the susceptible CML444 after 1, 6 and 16 hours. CKSBL10008 was less preferred than the susceptible CML444 after 1 and 6 hours only”. These sentences are written fairly. Please combine them into a single sentence.

-          Line 287: “CML444after”. Separate the words with a space.

-          Figure 2: Delete the sentence "Percentage of neonates on each leaf portion" in the upper part of the figure since it is already indicated in the title.

-          Figure 5: Delete the sentence “Percentage of neonates on each silks portion” in the upper part of the figure since it is already indicated in the title.

-          Figure 6: Delete the sentence “Percentage of neonates per ear” in the upper part of the figure since it is already indicated in the title.

Discussion:

-          Many "key" and recent references were missed in the discussion section:

o   Kasoma, C.; Shimelis, H.; Laing, M.D.; Mekonnen, B. Fall Armyworm Infestation and Development: Screening Tropical Maize Genotypes for Resistance in Zambia. Insects 2022, 13, 1020, doi:10.3390/insects13111020.

o   López-Castillo, L.M.; Silva-Fernández, S.E.; Winkler, R.; Bergvinson, D.J.; Arnason, J.T.; García-Lara, S. Postharvest Insect Resistance in Maize. J. Stored Prod. Res. 2018, 77, 66–76, doi:10.1016/j.jspr.2018.03.004.

o   Matova, P.M.; Kamutando, C.N.; Kutywayo, D.; Magorokosho, C.; Labuschagne, M. Fall Armyworm Tolerance of Maize Parental Lines, Experimental Hybrids, and Commercial Cultivars in Southern Africa. Agronomy 2022, 12, 1463, doi:10.3390/agronomy12061463.

o   Mukanga, M.; Matumba, L.; Makwenda, B.; Alfred, S.; Sakala, W.; Kanenga, K.; Chancellor, T.; Mugabe, J.; Bennett, B. Participatory Evaluation of Groundnut Planting Methods for Pre-Harvest Aflatoxin Management in Eastern Province of Zambia. Cah. Agric. 2019, 28, 1, doi:10.1051/cagri/2019002.

o   Santos, L.M.; Redaelli, L.R.; Diefenbach, L.M.G.; Efrom, C.F.S. Larval and Pupal Stage of Spodoptera Frugiperda (J. E. Smith) (Lepidoptera: Noctuidae) in Sweet and Field Corn Genotypes. Braz. J. Biol. Rev. Brasleira Biol. 2003, 63, 627–633, doi:10.1590/s1519-69842003000400009.

o   Hafeez, M.; Li, X.; Ullah, F.; Zhang, Z.; Zhang, J.; Huang, J.; Chen, L.; Siddiqui, J.A.; Ren, X.; Zhou, S.; et al. Characterization of Indoxacarb Resistance in the Fall Armyworm: Selection, Inheritance, Cross-Resistance, Possible Biochemical Mechanisms, and Fitness Costs. Biology 2022, 11, 1718, doi:10.3390/biology11121718.

o   Prasanna, B.M.; Bruce, A.; Beyene, Y.; Makumbi, D.; Gowda, M.; Asim, M.; Martinelli, S.; Head, G.P.; Parimi, S. Host Plant Resistance for Fall Armyworm Management in Maize: Relevance, Status and Prospects in Africa and Asia. Theor. Appl. Genet. 2022, 135, 3897–3916, doi:10.1007/s00122-022-04073-4.

-          In addition, I strongly recommend rewriting one or two paragraphs in the discussion section to better understand and express the significance of the findings in relation to what was previously known about the chemical composition and some morphological characteristics of the maize lines. Here are some related references:

o   Yasoob, H.; Abbas, N.; Li, Y.; Zhang, Y. Selection for Resistance, Life History Traits and the Biochemical Mechanism of Resistance to Thiamethoxam in the Maize Armyworm, Mythimna Separata (Lepidoptera: Noctuidae). Phytoparasitica 2018, 46, 627–634, doi:10.1007/s12600-018-0692-4.

o   Rocandio-Rodríguez, M.; Torres-Castillo, J.A.; Juárez-Aragón, M.C.; Chacón-Hernández, J.C.; Moreno-Ramírez, Y. del R.; Mora-Ravelo, S.G.; Delgado-Martínez, R.; Hernández-Juárez, A.; Heinz-Castro, R.T.Q.; Reyes-Zepeda, F. Evaluation of Resistance of Eleven Maize Races (Zea Mays L.) to the Red Spider Mite (Tetranychus Merganser, Boudreaux). Plants 2022, 11, 1414, doi:10.3390/plants11111414.

o   Singh, G.M.; Xu, J.; Schaefer, D.; Day, R.; Wang, Z.; Zhang, F. Maize Diversity for Fall Armyworm Resistance in a Warming World. Crop Sci. 2022, 62, 1–19, doi:10.1002/csc2.20649.

o   Kim, E.Y.; Jung, J.K.; Kim, I.H.; Kim, Y. Chymotrypsin Is a Molecular Target of Insect Resistance of Three Corn Varieties against the Asian Corn Borer, Ostrinia Furnacalis. PLOS ONE 2022, 17, e0266751, doi:10.1371/journal.pone.0266751.

References

The following reference is not found:

-          20. Dinno, A. Dunn's Test of Multiple Comparisons Using Rank Sums. Package dunn.test 2017. https://cran.r-project.org/web/packages/dunn.test/dunn.test.pdf

Author Response

Kindly find our replies in red below.

Abstract

-          Line 20: “first reported in West Africa in 2016”. Please correct the sentence as indicated in the introduction section (Line 45; “…first reported in West and Central Africa in 2016 [2]”).

Done

-          Add quantitative results to this section.

Done

-          Improve on the conclusions of the abstract.

Done

Introduction

-          Lines 51-53: “Most farmers in African countries are resource-constrained smallholders and usually face problems in effectively controlling insect pests like stem borers and FAW using insecticides”. Add a reference for this statement.

Done

-          Lines 59-60: “several maize inbred lines resistant to S. frugiperda have been developed namely Mp708, Mp78:518 [6,7] and Mp716 [8]”. Several or three inbred lines? Please correct.

We corrected by “some maize inbred lines resistant to S. frugiperda have been developed like Mp708, Mp78:518 [6,7] and Mp716 [8]”.

Materials and methods

-          Line 85: Stick to either “FAW” or “S. frugiperda” through the text.

Considered through the text.

-          “In addition, ears at the R1 silking stage (7 days after silking) and at R3 milk stage were used”. What were the ears used for? Please complete the sentence.

Sentence completed by “for all laboratory experiments”.

-          Line 106: delete “Kenya Agriculture and Livestock Research Organization (KALRO)” and only keep “KALRO” to avoid the redundancy in the text.

Done

-          Lines 107-109: “Before planting, Emamectin benzoate (19 g/l) insecticide was sprayed on the ground and walls of the net house against insects that might interact with the experiment”. Indicate when the pesticide will be applied.

As indicated now in the sentence, the pesticide was applied one week before planting.

-          Line 111: “10 g of phosphate (DAP) were…”. Delete “phosphate” and keep “DAP” to avoid the repetition in the text.

Deleted

-          Lines 132-133: “To estimate the sex ratio, approximately 200 neonate larvae from the hatched eggs of each female from each inbred line were reared”. Why is it around 200?  Please explain.

A minimum number of 200 neonates (randomly selected) giving a chance to represent the sex ratio of a progeny.

-          Lines 141-143: “RGRs were calculated by subtracting the average weight per larva on each inbred line at infestation from the average weight per larva on the same inbred line at the day of weighing, divided by the number of days in between”. This lengthy sentence should be replaced with a formula, in my opinion. Also, include the formulae for survival%, egg hatch%, adult emergence%, and so on.  

Changed as requested.

-          Lines 148-149: “On shelled ear and silks, only RGR after 7 and 10 days of feeding respectively were assessed”. Explain why only The RGR was evaluated. How about the other tested parameters? 

RGR was showed to be the most significant parameter between lines for plant’s leaves, see results.

-          Line 150: 2 Replace “(Light: Dark)” by (L: D).

Done

-          Line 152: “2.6. Antibiosis assessment under net house conditions”. This section is unclear and should be rewritten in terms of experimental design and trial establishment (number of tested plants, plots, and replicates; ears and silk experiments?). 

All details on the experimental design and trial establishment (number of tested plants, plots) are explained and mentioned at sub-section 2.4 Plants used in the net house experiment. A figure was added to show this net house design.

-          Line 155-158: Explain why each plant received 5 neonate larvae while the silk and ear experiments received only 2 neonate larvae.

Explanations now given in the text: for leaves five neonate larvae = the minimum number of infestation to see differences between susceptible and resistant maize inbred lines (personal observations); for silks and ears two neonate larvae = the maximum number of infestation for silks and ears to avoid to be completely eaten after 14 days of infestation (personal observations).

-          Line 156: Indicate the properties of the plastic sheath used in the experiment.

The property is given in the sentence as follow: “An open plastic sheaths filled with water were placed around each plot of Figure 1 to minimize larval movements between different plots or inbred lines.”

-          Line 160: Indicate the properties of the net bag used in the experiment.

Also the property is mentioned in the sentence as follow: “Silks or ears were covered with a small net bag upon infestation to minimize larval escape from these plant’s organs.”

-          Line 165: “2.7. Antixenosis assessment under laboratory conditions”. Why not conduct the feeding preference tests in the field (e.g., in cages)? These tests will be used to validate the laboratory results.

Testing larval preference in the field or in cages of crawling larvae is not easy particularly for small larvae such as neonates. Moreover, this kind of protocol was already well validated in the literature using the protocol of Rharrabe et al. (Frontiers in Ecology and Evolution 2014) and de Fouchier et al. (Frontiers in Behavioral Neuroscience 2018).

-          Line 173: “Ten neonate larvae were released in the center of each dish”. FAW neonate larvae are cannibalistic, as we all know. Did you study the intraspecific competition of FAW larvae in the laboratory? Before performing the multiple-choice experiments, this parameter should be considered.

Actually, the neonates of FAW are not cannibalistic, see paper of Sokame et al. 2022 (Journal of Pest Science; https://doi.org/10.1007/s10340-022-01572-7). During the timing of these experiments with a maximum of 16 hours there is no risk of cannibalistic phenomenon and then no risk of intraspecific competition of FAW.

-          Line 179: Explain why the experiment was repeated 24 and 80 times for the binary and multiple-choice experiments, respectively.

For binary, due to the use of all possible combination a replication of 24 was the maximum necessary to see differences between combinations.

-          I have some reservations about the statistical analyses. This section should be rewritten and divided into sub-titles to make each test easier to understand.

Sub-titles have been inserted to make easier to understand.

-          Lines 192-193: “the Wilcoxon rank sum test and Kruskal-Wallis rank sum test using dunn.test package version 1.3.5”. Please rewrite the sentence to avoid any confusion.

Done

-          Line 194: “Statistical significance was defined at the 5% level for all comparisons”. Delete this sentence.

Deleted

Results: 

-          Table 1: "(number of females/total number)" should be removed from the title.

Done

-          Line 224: Replace “the relative growth rates (RGR)” by “RGR”.

Done

-          Line 246: Replace “line” by “lines”.

Done

-          Lines 266-268: “Mp716 was less preferred than CML444 after 1, 6 and 16 hours. CML370 was also less preferred than the susceptible CML444 after 1, 6 and 16 hours. CKSBL10008 was less preferred than the susceptible CML444 after 1 and 6 hours only”. These sentences are written fairly. Please combine them into a single sentence.

Modified

-          Line 287: “CML444after”. Separate the words with a space.

Done

-          Figure 2: Delete the sentence "Percentage of neonates on each leaf portion" in the upper part of the figure since it is already indicated in the title. 

Done

-          Figure 5: Delete the sentence “Percentage of neonates on each silks portion” in the upper part of the figure since it is already indicated in the title.

Done

-          Figure 6: Delete the sentence “Percentage of neonates per ear” in the upper part of the figure since it is already indicated in the title.

Done

Discussion: 

-          Many "key" and recent references were missed in the discussion section:

We targeted papers that are directly linked to the main results that we have obtained.

Kasoma, C.; Shimelis, H.; Laing, M.D.; Mekonnen, B. Fall Armyworm Infestation and Development: Screening Tropical Maize Genotypes for Resistance in Zambia. Insects 2022, 13, 1020, doi:10.3390/insects13111020.

This paper is mostly linked to the development of a method to rear FAW. This was not the main purpose of our paper since we used a rearing method that was already optimized by CIMMYT.

López-Castillo, L.M.; Silva-Fernández, S.E.; Winkler, R.; Bergvinson, D.J.; Arnason, J.T.; García-Lara, S. Postharvest Insect Resistance in Maize. J. Stored Prod. Res. 2018, 77, 66–76, doi:10.1016/j.jspr.2018.03.004.

This paper refers on the principal factors involved in maize kernel postharvest pest resistance. In our case, our results indicated that the maize kernels are not source of plant resistance as compared to leaves. This is the reason why this paper was not cited.

Matova, P.M.; Kamutando, C.N.; Kutywayo, D.; Magorokosho, C.; Labuschagne, M. Fall Armyworm Tolerance of Maize Parental Lines, Experimental Hybrids, and Commercial Cultivars in Southern Africa. Agronomy 2022, 12, 1463, doi:10.3390/agronomy12061463.

Now cited as local and exotic lines with FAW tolerance/resistance contribute to FAW resistance breeding program.

Mukanga, M.; Matumba, L.; Makwenda, B.; Alfred, S.; Sakala, W.; Kanenga, K.; Chancellor, T.; Mugabe, J.; Bennett, B. Participatory Evaluation of Groundnut Planting Methods for Pre-Harvest Aflatoxin Management in Eastern Province of Zambia. Cah. Agric. 2019, 28, 1, doi:10.1051/cagri/2019002.

This paper is on aflatoxin and groundnut management then we feel that this paper is out of context of our study, then not cited.

Santos, L.M.; Redaelli, L.R.; Diefenbach, L.M.G.; Efrom, C.F.S. Larval and Pupal Stage of Spodoptera Frugiperda (J. E. Smith) (Lepidoptera: Noctuidae) in Sweet and Field Corn Genotypes. Braz. J. Biol. Rev. Brasleira Biol. 2003, 63, 627–633, doi:10.1590/s1519-69842003000400009.

Now cited since our methodology was similar.

Hafeez, M.; Li, X.; Ullah, F.; Zhang, Z.; Zhang, J.; Huang, J.; Chen, L.; Siddiqui, J.A.; Ren, X.; Zhou, S.; et al. Characterization of Indoxacarb Resistance in the Fall Armyworm: Selection, Inheritance, Cross-Resistance, Possible Biochemical Mechanisms, and Fitness Costs. Biology 2022, 11, 1718, doi:10.3390/biology11121718.

The paper is linked to insect’s resistance to pesticide and then we feel that this paper is out of context of our study, then not cited. We avoid discussing on the insect’s resistance mechanisms towards maize plants chemistry but focusing only on the reverse.

Prasanna, B.M.; Bruce, A.; Beyene, Y.; Makumbi, D.; Gowda, M.; Asim, M.; Martinelli, S.; Head, G.P.; Parimi, S. Host Plant Resistance for Fall Armyworm Management in Maize: Relevance, Status and Prospects in Africa and Asia. Theor. Appl. Genet. 2022, 135, 3897–3916, doi:10.1007/s00122-022-04073-4.

This paper was already cited before our revision.

-          In addition, I strongly recommend rewriting one or two paragraphs in the discussion section to better understand and express the significance of the findings in relation to what was previously known about the chemical composition and some morphological characteristics of the maize lines. Here are some related references:

Yasoob, H.; Abbas, N.; Li, Y.; Zhang, Y. Selection for Resistance, Life History Traits and the Biochemical Mechanism of Resistance to Thiamethoxam in the Maize Armyworm, Mythimna Separata (Lepidoptera: Noctuidae). Phytoparasitica 2018, 46, 627–634, doi:10.1007/s12600-018-0692-4.

The paper is linked to insect’s resistance to pesticide and then we feel that this paper is out of context of our study, then not cited. We avoid discussing on the insect’s resistance mechanisms towards maize plants chemistry but focusing only on the reverse.

Rocandio-Rodríguez, M.; Torres-Castillo, J.A.; Juárez-Aragón, M.C.; Chacón-Hernández, J.C.; Moreno-Ramírez, Y. del R.; Mora-Ravelo, S.G.; Delgado-Martínez, R.; Hernández-Juárez, A.; Heinz-Castro, R.T.Q.; Reyes-Zepeda, F. Evaluation of Resistance of Eleven Maize Races (Zea Mays L.) to the Red Spider Mite (Tetranychus Merganser, Boudreaux). Plants 2022, 11, 1414, doi:10.3390/plants11111414.

Singh, G.M.; Xu, J.; Schaefer, D.; Day, R.; Wang, Z.; Zhang, F. Maize Diversity for Fall Armyworm Resistance in a Warming World. Crop Sci. 2022, 62, 1–19, doi:10.1002/csc2.20649.

Kim, E.Y.; Jung, J.K.; Kim, I.H.; Kim, Y. Chymotrypsin Is a Molecular Target of Insect Resistance of Three Corn Varieties against the Asian Corn Borer, Ostrinia Furnacalis. PLOS ONE 2022, 17, e0266751, doi:10.1371/journal.pone.0266751.

These three papers are now cited and a rewritten paragraph was done as recommended by the reviewer.

References

The following reference is not found:

-          20. Dinno, A. Dunn's Test of Multiple Comparisons Using Rank Sums. Package dunn.test 2017. https://cran.r-project.org/web/packages/dunn.test/dunn.test.pdf

This reference was well cited and also mentioned in the reference list.